# The Association of Active Living Environments and Mental Health: A Canadian Epidemiological Analysis

**DOI:** 10.3390/ijerph17061910

**Published:** 2020-03-15

**Authors:** Aysha Lukmanji, Jeanne V.A. Williams, Andrew G.M. Bulloch, Ashley K. Dores, Scott B. Patten

**Affiliations:** 1Department of Community Health Sciences, Cumming School of Medicine, University of Calgary, 3280 Hospital Drive NW Calgary, Calgary, AB T2N 4Z6, Canada; aysha.lukmanji1@ucalgary.ca (A.L.); akdores@ucalgary.ca (A.K.D.); 2Mathison Centre for Mental Health Research & Education, Hotchkiss Brain Institute, Cumming School of Medicine, University of Calgary, 3280 Hospital Drive NW Calgary, Calgary, AB T2N 4Z6, Canada; 3Cuthbertson & Fischer Chair in Pediatric Mental Health, University of Calgary, 3280 Hospital Drive NW Calgary, Calgary, AB T2N 4Z6, Canada

**Keywords:** built environments, mental health, active living environments, active transportation

## Abstract

Environments that promote use of active transport (walking, biking, and public transport use) are known as “active living environments” (ALE). Using a Canadian national sample, our aim was to determine if ALEs were associated with mental health outcomes, including depressive symptoms, and mood and anxiety disorders. Data from the Canadian Community Health Survey from 2015–2016 was used for demographic characteristics and mental health outcomes (*n* ≈ 110,000). This data was linked to datasets from the Canadian Urban Environmental Health Research Consortium, reporting ALE and social and material deprivation. Depressive symptoms were evaluated using standard dichotomized scores of 5+ (mild) and 10+ (moderate/severe) from the Patient Health Questionnaire-9. Self-reported diagnosed mood and anxiety disorders were also included. Logistic regression was used to determine the association of mental health outcomes with four classes of ALE. The analysis included adjustments for social and material deprivation, age, sex, chronic conditions, marital status, education, employment, income, BMI, and immigrant status. No association between any mental health outcome and ALE were observed. While the benefits of ALE to physical health are known, these results do not support the hypothesis that more favorable ALE and increased use of active transport is associated with better mental health outcomes.

## 1. Introduction

Elements of the built environment which promote physical activity are believed to be associated with positive physical health outcomes. Research has established that features of the built environment, including intersections, points of interest, dwelling density, and transit availability are linked to an increased type of activity classified as “active transportation” [1]. Active transportation is defined by participation in activities such as walking and cycling, combined with use of public transportation, as a means for arriving at a particular destination. “Active living environments” are regions which promote the use of active transport due to features of the built environment. Increased physical activity from living in an environment conducive to being active has been associated with a lower risk of obesity among other chronic conditions, such as diabetes and cardiovascular disease [2,3,4,5]. Despite the increasing number of studies demonstrating the association between physical health and active transportation, interest towards the impact of these active environments on mental health outcomes is increasing.

Since the 1990′s, depressive disorders have been recognized as a leading global burden of disease, making treatment and prevention a primary public health concern [6]. Major Depressive Disorder (MDD) accounted for 7.5% of years lived with disability, and is ranked as the single largest contributor to global disability [6]. Therefore, finding cost-effective public health interventions for MDD is a priority worldwide. One such intervention, exercise, has been hypothesized to improve symptoms of depression through several mechanisms, including the release of endocannabinoids, endorphins, increasing social cohesion, and building self-confidence. Systematic reviews have indicated that physical activity may reduce the risk of depression; however, few have focused on activities exclusively linked to the built environment [7,8]. Studies which assess the relationship between neighborhoods and mental health often focus more on aesthetic characteristics, like greenness, parks, or access to specific facilities [2].

Of the few previous studies which specifically assess the relationship between neighborhood characteristics promoting active transportation and mental health outcomes, results are inconsistent. Negative findings from previously published longitudinal studies may have lacked power to detect an expectedly small association [9]. Living in an environment which promotes use of active transportation is one of numerous factors which may impact an individual’s mental health. Therefore, to observe an association between mental health and active living environments, a large epidemiological sample size is likely to be necessary to quantify the effect.

Much of the previous research conducted on the relationship between physical activity promoted by the environment and mental health have either focused exclusively on recreational physical activity, or have mixed recreational and non-recreational activity [2]. Therefore, although various systematic reviews may demonstrate a relationship between physical activity and mental health, the definition of activity tends to be heterogeneous [10]. Since there are multiple hypothesized mechanisms through which physical activity may promote mental health, it may similarly be hypothesized that not all types of physical activity have the same ability to promote positive mental health. Therefore, it may be important to differentiate the type of physical activity, particularly recreational and non-recreational types of activities.

Supported by previous research, we hypothesize that people residing in better active living environments and who specifically take more part in active transportation will have better mental health outcomes. Using a Canadian epidemiological sample, our objective is to determine whether active living environments and the use of active transportation are associated with more positive indicators of mental health.

## 2. Materials and Methods

### 2.1. Study Population (Canadian Community Health Survey)

Analysis was conducted using cross-sectional survey data from the Canadian Community Health Survey (CCHS) during the years 2015 and 2016 [11]. The target population of the CCHS included members of the Canadian population aged 12 and over, but excluded those living on Aboriginal settlements, who were in foster care, who were full-time members of the Canadian Armed Forces, or who were part of the institutionalized population and the remote Région du Nunavik or Région des Terres-Cries-de-la-Baie-James. Excluded populations only accounted for 3% of the entire Canadian population; therefore, the CCHS is representative at the provincial and health region level [11]. Individual households were first selected to participate in the CCHS, then by using various selection probabilities, individuals from those households were randomly selected. The combined response rate (the product of household and individual response rates) for 2015 and 2016 CCHS was 59.5%.

The CCHS collected information relating to health determinants, status, and health care utilization of the respondents. The core content of the survey was asked to all respondents each survey year, but “optional content” questions were reserved for provinces/territories who had opted to administer them. Interviews were conducted using computer-assisted personal and telephone interview software; therefore, surveys were designed by Statistics Canada to follow a logical flow. Post-survey results were assessed for quality and compared to previous years to prevent errors [11].

### 2.2. Health Outcomes

#### 2.2.1. Patient Health Questionnaire-9

The depression module of the CCHS was administered to selected provinces each survey year. In the combined years 2015 and 2016 of the CCHS, the depression module was administered to eight provinces/territories, including Newfoundland and Labrador, Prince Edward Island, Nova Scotia, New Brunswick, Ontario, Manitoba, Saskatchewan, and the Northwest Territories. During the depression module, respondents answered the Patient Health Questionnaire-9 (PHQ-9) to assess risk of a MDE [12]. The PHQ-9 also consists of nine items which align with the symptom-based criteria for Major Depressive Episode in recent editions of DSM-5. The nine items are assessed based on the past 2 weeks, and response options include “*not at all”, “several days”, “more than half the days”, or “nearly every day”,* corresponding with a score of 0–3, respectively. The final overall score range is 0–27, a total of ≥5 indicating at least mildly depressive symptoms, and ≥10 as moderate to severe symptoms [12].

#### 2.2.2. Self-Reported Mood and Anxiety Disorders

Self-report of mood and anxiety disorders was also included in the analysis. All respondents of the CCHS were asked to report if they had received a diagnosis of a mood or anxiety disorder from a “health professional” with separate items for mood and anxiety disorders.

#### 2.2.3. Social and Material Deprivation

Social and material deprivation were also expected to exhibit a spatial pattern. Where material deprivation is concerned with the lack of access to basic goods and amenities, social deprivation is the inadequacy of social networks in many forms, from family to the community [13]. Therefore, we obtained another database from CANUE that assessed social and material deprivation at the postal code level. Social and material deprivation were derived from the Pampalon, Hamel, Gamache, Philibert, Raymond and Simpson [13] index, and each postal code was assigned a quintile value, from least deprived (quintile 1) to most deprived (quintile 5) [1]. The index was developed using six indicators from census data, and standardized for the age and sex structure of the Canadian population. The specific indicators included were calculated using the population aged 15 years and older. These indicators include: the employment to population ratio, the average income of the population, the proportion of single-parent families, the proportion who were single, divorced, and widowed, proportion living alone, and proportion without a high school diploma or equivalent. The indicators for education, employment, and income were used to develop the material deprivation quintiles, while the indicators on marital status and family structure were used to develop the social deprivation quintiles, as confirmed by principal component analysis [13]. Although it is possible to combine indicators, social and material deprivation are different concepts and were included separately in the analysis.

#### 2.2.4. Body Mass Index

Since there is a well-established relationship between use of active transport and lower prevalence of obesity, models were also run with obesity as an outcome to confirm that the expected relationship with physical health would be found [3]. Respondents’ BMI should decrease as their environments become more conducive to use of active transport. BMI was dichotomized to assess the proportion of respondents who were classified as “obese” compared to all respondents. Obesity was classified using self-reported height and weight to calculate respondents’ BMI. Obesity was calculated using the international standards. The international standard for obesity corresponds to a BMI 30 or greater, for all ages over 12 [14,15].

### 2.3. Active Living Environments (Exposure)

The dataset on regional ALEs in 2016 was obtained through the Canadian Urban Environmental Health Research Consortium (CANUE) [16,17,18]. The database was developed to measure the “active living friendliness” of Canadian communities and to be linked to various national and local surveys including the CCHS. Potential indicators of ALE were first selected by a literature review, then various measures were narrowed down by considering factors such as data availability, association with physical activity, and ability to be measured across different geographic regions within Canada. Consequently, classes of ALEs were developed using combined validated measurements of intersection density, dwelling density, and points of interest (bars, schools, shops, restaurants, businesses, etc.) [19,20]. Also, the ALE dataset had been previously validated in a cross-Canada analysis, assessing the correlation of the ALE measure with walking to work and use of active transport [21].

Measures for regional ALEs were calculated using 1 km circular buffers around a centroid (middle point) of a dissemination area, which is the smallest area for which census data is disseminated in Canada. The ALE class score was developed using a cluster analysis which assigned each dissemination area a score of 1–5 based on the three aforementioned measures. Scores from 1–5 corresponded with very low, low, moderate, high, and very high ALE, respectively.

### 2.4. Statistical Analysis

Stata version 15.0 (StataCorp, College Station, TX, USA) was used to conduct all analyses [22]. Statistical significance was determined using α = 0.05, and confidence intervals were calculated at the 95% confidence level. Replicate bootstrap weights were provided by Statistics Canada to address the variation in selection probabilities of CCHS respondents and clustering which resulted from the design effects of a multi-stage sampling procedure. A set of 1000 replicate bootstrap weights were used for analysis to produce valid estimates and correct standard errors. Bootstrap weights were also adjusted by Statistics Canada in order to reduce the risk of selection bias arising from non-response. Both ALE and social and material deprivation datasets were merged with the CCHS using residential 6-digit postal codes.

Due to the limited number of respondents in each ALE class, classes 4 and 5 (indicating high and very high ALE) were combined to create a total of four classes. The association between ALEs and mental health was assessed using four measures: self-reported mood disorder, self-reported anxiety disorder, and overall PHQ-9 score dichotomized to ≥5 and ≥10. Since all outcomes were binary, logistic regression was used to model the association between ALEs and mental health. Models also included terms for potential confounders: age, sex, social and material deprivation, obesity, chronic conditions, marital status, highest level of education, and employment status. In the CCHS, body mass index (BMI) was used to assess the categories of underweight, normal, overweight, and obese. The outcome was dichotomized to compare respondents in the “obese” category to all others.

## 3. Results

The flow diagram Figure A1 in Appendix B describes the final number of respondents for each health outcome after merging the CCHS dataset with the CANUE ALE and social and material deprivation dataset. Table 1 describes the demographic characteristics of the overall sample and was stratified by ALE class. ALEs were intended to act as an environmental measure for use of active transport. Notably, not only did the proportion of people who reported using active transport increase from the worst to best class, but the mean number of minutes engaged in active transport also increased by 112% respectively (Table 1). The observed increase affirms that those in better ALE participate more in the use of active transport. Also, as demonstrated by Table 1, mean minutes of recreational activity are consistent at approximately 110 minutes for all ALE classes. Where the fifth quintiles represent the most social and material deprivation, results from Table 1 show that there is no clear trend of deprivation throughout the classes of ALE. Other demographic characteristics, for example, mean age and proportion married, remain relatively constant throughout the classes of ALE (Table 1).

Odds ratios (OR) and *p*-values from crude and adjusted models are depicted by Table 2 and Table 3 (full adjusted models available in Appendix A). Results using a PHQ-9 dichotomization of 5 are reported in Table 2 since they yielded similar results to analysis using the cut-off of 10. Results demonstrate the expected associations between mental health outcomes and social and material deprivation, with adjustment for sex, age, education, marital status, employment, income, obesity, and chronic conditions. In these models, material deprivation (measured using an environmental measure) was not consistently associated with mental health, but both income and education, the two main elements of individually assessed socio-economic status, were associated. The model does not demonstrate an effect of neighborhood material deprivation that is independent of individual-level socioeconomic status.

Both adjusted and crude models demonstrate that no association was found between ALE and any of the mental health outcomes measured (Table 2 and Table 3). The estimates presented in Table 2 and Table 3 are precise, with the upper confidence limits for the adjusted OR falling in the range of 1.05 to 1.54, and each of the 95% confidence intervals overlapping with an OR of 1. While the study cannot rule out weak effects, the results are sufficiently precise to be inconsistent with a strong effect. Despite the lack of significant findings for the mental health outcomes studied, the crude and adjusted models show that, as expected, respondents living in better ALE are less likely to be classified as obese (Table 2 and Table 3). Specifically, the adjusted model assessing obesity as an outcome yields an OR of 1.65 (95%CI 1.43–2.03) for those in the lowest ALE (class 1). In fact, both crude and adjusted models demonstrate a dose-response relationship between ALE and obesity, where the association is only slightly weaker in adjusted models compared to the crude. Interestingly, in the case of obesity, adjusted household income was not significant, whereas area-based material deprivation was.

## 4. Discussion

Overall, analysis did not demonstrate any association between measures of mental health and active living environments. However, the expected dose-response effect using obesity as an outcome was observed. Also, the relationship between social deprivation and a set of well-known determinants of mental health demonstrated the expected effects in the models. These results suggest that the relationship between the built environment and mental health differs from that of physical health.

ALEs may be connected to mental health through several mechanisms. Related factors to ALEs, such as physiological activity and pleasantness of the environment, may modify the effect of ALEs on mental health outcomes. Therefore, it may be hypothesized that unlike physical health, the perception of one’s environment may be more important for mental health outcomes than the amount of physical activity the environment stimulates. Previous research investigating the relationship between the built environment and mental health have reported a significant association of positive mental health with parks, greenness, and proximity to various services [9,23]. Therefore, it is possible that individuals who use active transportation do not benefit from the increased activity levels if they perceive the activity as not pleasurable or the environment as not aesthetically pleasing. This hypothesis is supported by a previous analysis using survey data in Australia [23]. In this analysis, perceived environmental greenness was associated with mental but not physical health of participants. Notably, the same analysis indicated that the relationship between perceived greenness and mental health was only partly accounted for by recreational walking and social coherence. Another analysis by Srugo, et al. [24] found no association between the objective measure of greenness surrounding schools and children’s mental health. These results support the hypothesis that perception may moderate the potential benefit of increased use of active transport or that such an effect is multi-causal. Since the level of greenness is not captured by the ALE index used here, and we were unable to stratify by level of greenness, the potential benefit of living in favorable ALEs may have been missed.

Similarly, associations between ALEs and mental health might be specific to cultural context and other environmental variables. In Canada, due to high latitudes, winters are longer compared to other regions closer to the equator. Climate may be an effect modifier of the association of living in an ALE and mental health, because certain weather events may cause the experience of using active transportation to be more onerous. In such a case, similar analyses conducted in different countries would produce opposing results.

Alternatively, it is possible that the amount of activity promoted by one’s environment is insufficient in quantity and/or rigor to have any effect on depressive symptoms or mood and anxiety disorders. The World Health Organization (WHO) advises adults to participate in at least 150 minutes of moderate-intensity aerobic physical activity per week to meet its guidelines [6]. As reported in Table 1, the mean minutes of active transportation use increases correspondingly to the ALEs, but only respondents residing in the best ALE meet the temporal requirements. Also, it was not possible to measure the intensity of activity related to active transport use, so even those meeting the minimum required minutes of activity may not be participating in sufficient aerobic rigor. In such a case, activity from recreation and other sources would be more beneficial in the context of WHO standards than use of active transport.

There were several strengths of this analysis, the main one being the quality and generalizability of the CCHS. Survey data from the CCHS can be generalized to the Canadian household population. Similarly, the epidemiological sample size of the CCHS helped to reduce to likelihood that results were negative merely due to the lack of power to detect an effect. Also, as presented in Table 1 and validated by Herrmann, Gleckner, Wasfi, Thierry, Kestens and Ross [21], the CANUE ALE classes have been validated to correspond with an increase of walking to work and use of active transportation.

Although the CCHS was epidemiological and representative of the Canadian household population, our analysis was not without limitations. The use of bootstrap weights to account for survey design effects is a major strength of the analysis, since the weighting procedure ensures estimates are accurately representative of the Canadian household population. However, the nature of the weights provided by Statistics Canada limits the ability to use a multilevel model, since they are only provided at one level. Fitting a multilevel model would be favorable, since such a model would account for the compositional and contextual effects in addition to cross-level interactions. Also, the CCHS is focused on healthcare-related topics; therefore, due to data availability, we were unable to control for various neighborhood characteristics. Since the survey is only representative of Canadians aged ≥12 years old, it was not possible to extrapolate results to other age groups. The cross-sectional nature of the data was limiting since respondents’ time residing at their postal code was not assessed. Consequently, it was not possible to determine how long respondents have resided in their respective ALE and if there were previous environmental stressors which may have impacted their mental health. During the survey years 2015–2016, information regarding respondents’ time residing at their perception of their environments and neighborhood crime rates were not collected, each of which could be important confounders. Despite being validated, the ALE did not include some common indicators of active living, such as traffic volume, crime rates, aesthetics, size and number of sidewalks, and topography [25]. Utilizing active transport may, in fact, have a negative impact on mental health if crime rates are high, or respondents perceive their environment as unsafe. Also, as previously noted, perception of one’s environment may modify the relationship between ALEs and mental health. Measurement of ALEs also presented various limitations. ALEs only evaluated whether a region promoted active transport or not; therefore, the health-promoting ability of points of interest were not valued differently from one another—for example, existence of a health food store would be comparable to that of a fast food chain. Previous research indicates that access to various services may produce increasingly positive mental health outcomes, particularly for individuals with chronic conditions. Consequently, it may be useful to assess various points of interest differently in the production of ALE classes.

## 5. Conclusions

In this study, we did not find evidence that ALEs are associated with better mental health. However, the expected associations between obesity and ALEs were observed. Future analyses should employ multilevel modelling procedures to account for contextual and compositional effects and their cross-level interactions. Further research is also required to determine if environmental perceptions moderate the effect of increased active transportation use on mental health. These future studies should focus on how the duration living in a given ALE and presence of previous environmental stressors might moderate the association between ALE and mental health.

## Figures and Tables

**Table 1 ijerph-17-01910-t001:** Proportion of demographic characteristics overall and by the active living environment class with 95% confidence intervals.

Demographic Characteristics	Overall CCHS (%) *n* ≈ 110,000	Active Living Environments			
		Class 1 *n* ≈ 49,000 (%) (Very Low)	Class 2 *n* ≈ 33,000 (%) (Low)	Class 3 *n* ≈ 19,000 (%) (Moderate)	Class 4/5 *n* ≈ 8000 (%) (High/ Very High)
Used Active Transport ^a^	46.9 (46.3, 47.4)	36.0 (35.2, 36.8)	44.4 (43.3, 45.5)	49.89 (48.7, 51.1)	68.87 (67.2, 70.6)
Active Transport ^a,b^ (mean minutes)	96.8 (94.58, 98.3)	73.2 (70.2, 76.1)	85.7 (82.5, 88.9)	99.9 (95.3, 104.6)	162.4 (153.3, 171.5)
Recreational Activity ^a^ (mean minutes)	111.1 (108.5, 113.7)	110.2 (106.5, 114.0)	118.2 (113.4, 122.9)	102.7 (98.0, 107.4)	111.9 (102.1, 121.7)
Hours spent sedentary					
0–14	26.0 (25.3, 26.8)	28.4 (27.2, 29.6)	24.8 (23.6, 26.0)	25.5 (24.0, 27.0)	25.8(23.6, 27.9)
15–24	25.5 (24.8, 26.2)	26.2 (25.0, 27.3)	25.7 (24.4, 26.9)	25.5 (24.2, 26.9)	24.0 (22.0, 26.0)
23–34	18.6 (18.0, 19.2)	18.1 (17.2, 19.0)	18.7 (17.6, 19.6)	18.4 (17.1, 19.7)	19.6 (17.8, 21.4)
32+	29.9 (29.2, 30.7)	27.4 (26.2, 28.5)	30.9 (29.7, 32.1)	30.6 (29.1, 32.1)	30.6 (28.4, 32.9)
Material Deprivation					
1 (Lowest)	20.9 (20.0, 21.8)	15.0 (13.5, 16.5)	23.9 (22.0, 25.8)	17.4 (15.3, 19.4)	32.9 (29.6, 36.1)
2	20.3 (19.3, 21.3)	22.9 (21.3, 24.4)	25.5 (23.4, 27.5)	14.8 (12.8, 16.7)	13.0 (10.2, 15.9)
3	20.6 (19.5, 21.3)	22.9 (21.3, 24.5)	21.3 (19.4, 23.1)	21.5 (19.1, 24.0)	12.8 (9.9, 15.7)
4	19.8 (18.8, 20.8)	21.8 (20.3, 23.3)	17.7 (16.0, 19.5)	21.5 (19.1, 23.9)	17.5 (14.4, 20.5)
5 (Highest)	18.4 (17.5, 19.3)	17.4 (16.1, 18.7)	11.7 (10.2, 13.2)	24.9 (22.5, 27.3)	23.8 (20.5, 27.1)
Social Deprivation					
1 (Lowest)	18.3 (17.4, 19.2)	19.2 (17.8, 20.7)	23.0 (21.1, 25.0)	19.8 (17.4, 22.2)	3.4 (2.0, 4.8)
2	18.5 (17.5, 19.4)	26.3 (24.6, 27.9)	19.2 (17.4, 21.0)	14.4 (12.3, 16.5)	7.4 (5.1, 9.7)
3	19.6 (18.6, 20.6)	27.2 (25.5, 28.9)	19.2 (17.4, 21.0)	14.3 (12.3, 16.3)	13.4 (10.6, 16.2)
4	21.3 (20.3, 22.4)	19.4 (17.9, 20.8)	19.8 (18.0, 21.5)	21.3 (19.0, 23.6)	28.7(25.0, 32.5)
5 (Highest)	22.4 (21.4, 23.3)	7.9 (7.0, 8.9)	18.8 (17.3, 20.4)	30.2 (28.0, 32.5)	47.0 (43.1, 50.9)
Age (Mean)	45.3 (45.2, 45.4)	46.8 (46.5, 47.1)	45.2 (44.9, 45.50)	44.4 (44.0, 44.8)	42.0 (43.4, 44.5)
Sex					
Female	50.7 (50.6, 50.7)	49.9 (49.3, 50.4)	50.0 (49.3, 50.8)	51.7 (50.7, 52.6)	52.1 (51.0, 53.1)
Marital status					
Married/Common Law	57.8 (57.3, 58.3)	63.6 (62.9, 64.3)	59.6 (58.7, 60.6)	53.4 (52.3, 54.6)	48.9 (47.1, 50.8)
Single	30.1 (29.7, 30.5)	25.3 (24.7, 25.9)	28.7 (27.9, 29.6)	33.2 (32.1, 34.2)	37.6 (36.0, 39.2)
Widowed/ separated/ divorced	12.2 (11.8, 12.5)	11.1 (10.7, 11.6)	11.7 (11.1, 12.2)	13.4 (12.7, 14.1)	13.5 (12.4, 14.6)
Highest Level of Education					
Less Than Secondary School	18.5 (18.2, 18.8)	22.2 (21.5, 22.8)	17.3 (16.7, 18.0)	18.1 (17.2, 19.0)	13.6 (12.5, 14.7)
No Post-Secondary	23.2 (22.8, 23.7)	24.4 (23.7, 25.1)	23.4 (22.6, 24.2)	23.8 (22.8, 24.7)	19.5 (18.0, 21.0)
Certificate, Diploma, or University Degree	58.3 (57.8, 58.8)	53.4 (52.6, 54.3)	59.3 (58.3, 60.3)	58.1 (57.0, 59.3)	66.9 (65.1, 68.6)
Employment Status (last week)					
Worked	61.9 (61.4, 62.4)	60.3 (59.7, 61.0)	62.8 (61.8, 63.7)	61.8 (60.6, 63.0)	63.4 (61.6, 65.1)
Absent from job	4.9 (4.6, 5.2)	5.3 (4.9, 5.7)	5.3 (4.8, 5.7)	4.3 (3.8, 4.8)	4.2 (3.4, 5.0)
No job	33.2 (32.7, 33.7)	34.4 (33.7, 35.1)	31.9 (31.0, 32.9)	33.9 (32.7, 35.0)	32.5 (30.7, 34.2)
Adjusted Household Income (Ratio to Low Income Cut-off)					
Quartile 1 (Highest)	26.0 (25.6, 26.5)	19.5 (19.0, 20.1)	24.0 (23.1, 24.8)	30.7 (29.6, 31.8)	36.1 (34.5, 37.8)
Quartile 2	25.2 (24.8, 25.7)	25.6 (25.0, 26.2)	25.9 (25.1, 26.7)	24.7(23.6, 25.7)	24.1 (22.6, 25.5)
Quartile 3	24.2 (23.7, 24.6)	25.6 (25.0, 26.2)	23.3 (22.6, 24.1)	23.9 (23.0, 24.8)	23.6 (22.2, 25.0)
Quartile 4 (Lowest)	24.6 (24.2, 25.0)	29.3 (28.7, 29.9)	26.8 (26.0, 27.6)	20.7 (19.8, 21.7)	16.2 (14.9, 17.6)
Landed immigrant, non-permanent resident	25.7 (25.2, 26.3)	9.9 (9.3, 10.6)	23.0 (21.7, 24.2)	37.3 (35.8, 38.9)	45.7 (43.65, 47.8)
Current Smoker	17.4 (16.9, 17.8)	18.2 (17.6, 18.8)	16.4 (15.7, 17.1)	17.1 (16.2, 18.0)	17.7 (16.3, 19.1)
Binge Drinker ^c^	25.0 (24.4, 25.5)	27.0 (26.3, 27.8)	23.5 (22.6, 24.4)	23.9 (22.7, 25.0)	25.5 (23.6, 27.5)
BMI- Standard International					
Obese	19.1 (18.7,19.6)	22.2 (21.5, 22.8)	19.8 (19.0, 20.6)	17.8 (16.9, 18.7)	13.4 (12.0, 14.5)
Chronic Conditions					
Has at least 1 of 10 chronic conditions ^d^	50.7 (50.1, 51.2)	53.4 (52.6, 54.2)	51.2 (50.2, 52.1)	49.9 (48.8, 51.0)	45.1 (43.4, 46.9)

^a^ Only asked to respondents age 18+; ^b^ Drops respondents who answered >1800 minutes of active transport a week, accounting for less than approximately 1% of the sample. ^c^ Five or more drinks for males and 4+ for female on more than one occasion over one month in the past year; ^d^ Includes: Arthritis, asthma, bronchitis, COPD, back problems, diabetes, heart disease, high blood pressure, stroke, migraine.

**Table 2 ijerph-17-01910-t002:** Crude models of mental health outcomes and obesity.

Exposure	PHQ-9 10+		Mood Disorder		Anxiety Disorder		Obesity	
**ALE**	OR (95% CI)	*p*-value	OR (95% CI)	*p*-value	OR (95% CI)	*p*-value	OR (95% CI)	*p*-value
Class 1	0.82 (0.64–1.05)	0.12	0.94 (0.82, 1.07)	0.36	1.08 (0.95, 1.23)	0.24	1.86 (1.66–2.08)	<0.01
Class 2	0.80 (0.62–1.04)	0.10	1.00 (0.88, 1.51)	0.90	1.08 (0.95, 1.23)	0.25	1.61 (1.44–1.82)	<0.01
Class 3	0.97 (0.74–1.27)	0.81	1.02 (0.89, 1.17)	0.78	1.14 (0.99, 1.30)	0.06	1.42 (1.26–2.61)	<0.01
Class 4/5	ref		ref		ref		ref	

**Table 3 ijerph-17-01910-t003:** Adjusted odds of mental health outcomes and obesity *.

	PHQ-9 10+		Mood Disorder		Anxiety Disorder		Obesity	
Exposure	aOR (95% CI)	*p*-value	aOR (95% CI)	*p*-value	aOR (95% CI)	*p*-value	aOR (95% CI)	*p*-value
Active Living Environment								
Class 1	0.98 (0.71–1.35)	0.90	0.88 (0.74, 1.05)	0.16	0.95 (0.80, 1.13)	0.56	1.65 (1.43–2.03)	<0.01
Class 2	0.92 (0.67–1.25)	0.59	0.93 (0.79, 1.09)	0.36	0.98 (0.84, 1.16)	0.84	1.47 (1.20–1.81)	<0.01
Class 3	1.11 (0.80–1.54)	0.53	0.95 (0.80, 1.12)	0.51	1.01 (0.85, 1.19)	0.94	1.21 (0.97–1.50)	0.09
Class 4/5	ref		ref		ref		ref	
Material Deprivation								
Quintile 1	ref		ref		ref		ref	
Quintile 2	0.93 (0.72–1.19)	0.56	0.89 (0.78, 1.01)	0.07	1.07 (0.93, 1.22)	0.34	1.32 (1.14–1.53)	<0.01
Quintile 3	0.85 (0.67–1.07)	0.15	1.04 (0.92, 1.19)	0.52	1.17 (1.02, 1.34)	0.02	1.32 (1.15–1.52)	<0.01
Quintile 4	0.95 (0.74–1.22)	0.71	1.01 (0.89, 1.15)	0.82	1.30 (1.13, 1.49)	<0.01	1.68 (1.45–1.94)	<0.01
Quintile 5	1.00 (0.78–1.28)	1.00	0.93 (0.82, 1.07)	0.32	1.39 (1.22, 1.59)	<0.01	1.83 (1.57–2.13)	<0.01
Social Deprivation								
Quintile 1	ref		ref		ref		ref	
Quintile 2	1.37 (1.07–1.77)	0.01	1.15 (0.99, 1.34)	0.07	1.16 (0.98, 1.37)	0.08	1.20 (1.05–1.37)	<0.01
Quintile 3	1.51 (1.17–1.93)	<0.01	1.14 (0.98, 1.33)	0.09	1.22 (1.04, 1.43)	0.02	1.26 (1.11–1.44)	<0.01
Quintile 4	1.61 (1.29–2.01)	<0.01	1.25 (1.08, 1.45)	<0.01	1.20 (1.03, 1.40)	0.02	1.25 (1.10–1.42)	<0.01
Quintile 5	2.05 (1.61–2.59)	<0.01	1.50 (1.29, 1.75)	<0.01	1.45 (1.24, 1.69)	<0.01	1.23 (1.07–1.40)	<0.01

* adjusted for chronic conditions age, sex, marital status, highest level of education, employment status, income quartile, immigrant status, and obesity when obesity is not an outcome.

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
