# Peer review of "The Association of Active Living Environments and Mental Health: A Canadian Epidemiological Analysis"

_ijerph, 2020, doi:10.3390/ijerph17061910_

Round 1

Reviewer 1 Report

Manuscript ID: ijerph-738806

Title: The Association of Active Living Environments and Mental Health:

A Canadian Epidemiological Analysis

The present manuscript comprise results on the association of active living environments and three mental health outcomes as well as obesity in a quite large Canadian survey. The authors present an interesting analysis on mental health which was aligned with the analysis of obesity as a well-known association with the built environment. Thus, I support the publication of this analysis, although it did not reveal an association with mental health in this large sample, but actually adds to the body of literature on this widely assumed topic.

However, some major aspects of this manuscript should be clarified in the context of the literature and the presented analysis.

  1. The authors very specifically focus on the link between opportunities for active transport in the built environment around the participants’ home neighbourhood and mental health. On the one hand the effect of the built environment on mental health is assumed to proceed through participation in more or less light physical activity, but not on other assumed links via perception of greenness (or blue spaces, i.e. lakes and rivers) and social cohesion in the neighbourhood which were or could not be considered. On the other hand, the single focus on ALEs reduces the dimensions of physical activity itself ignoring leisure time or high intensity PA. Since this might be difficult to add to the analysis on the exposure side which was also discussed by the authors, a more precise presentation of the literature and the focus of the analysis should be given in the background section (line 46 to 56).
  2. The different measures that were used to assess mental health in the study sample were sufficiently described, however the paragraph on BMI is confusing and presents more of a background information than the actual assessment of BMI (self-reported or measured weight and height? Which type of scale? Which cut-offs for 12 – 18 years old? Which bmi values were used as cut-off?). The authors might add this paragraph to the background, but should provide detailed information on the assessment which is confusingly given in line 146 to 148.
  3. The statistical analysis is not sufficiently described. The authors should first state the level of significance and the calculation of 95% confidence limits in this section. Further, please change ‘categorical’ (line 143) to binary, since no multinomial logistics regression was conducted.
  4. In the results section, results on other covariates are presented that are not revealed in the Tables (line 174 and 175). I recommend on providing a full table of the regression results as a supplement. However, to present results on an allegedly protective effect of immigrant status, which is then not discussed further, should be avoided. Please omit this sentence from the results section.
  5. Although the authors thoroughly discuss missing information to capture other dimensions of the built environment, it is important to note that a major limitation is induced through the age range and different residential biographies. Thus a strong bias is introduced by assuming only a cross-sectional short-term effect of the built environment on mental health, although it is unclear for how long participants were actually living in the assigned neighbourhood and if other environmental exposures from past neighbourhoods cumulatively lead to mental health problems. This should be discussed in more detail providing some insight on future research questions and changes in study design.
  6. The manuscript included some major spelling mistakes. The authors should again check the manuscript again and also consider past tense consistently throughout the manuscript.

Line 112: “use of active transport lower obesity,…”

Line 124: “… classes ALEs where developed…”

Line 178: “The models did not find…”

Line 225: due to high latitudes, winter are longer compared to regions closer to the equator.”

Line 226: “Climate may modify the effects of living in an ALE and mental health”

Reviewer 2 Report

Overall speaking, the paper is well-structured and written. The research design is clearly explained and findings are clearly presented. I just have a few concerns among the variables in the empirical study:

(1) The authors said data the ALE indicators were chosen by considering data availability, association with physical activity, and ability to be measured across different geographic regions, within Canada. Was there any validity of these indicators? Were these indicators particularly relevant to the concept "ALE"?

(2) What are the popular ALE indicators excluded in this study because of data unavailability? If yes, these should be named in the discussion section.

(3) More clear definitions of "Social Deprivation" and "Material Deprivation" are needed.

(4) What were the items or indicators used to gauge "Social Deprivation" and "Material Deprivation"?

Reviewer 3 Report

Interesting and important paper!

The authors have made a very strong claim about the effect of ecological factors on individual's outcomes based on the results obtained from a logistic regression analysis. Therefore, we are left with the following questions that cannot be answered by an OLS regression analysis:

  1. what about compositional effect (when only characteristics of the individuals are responsible for inter-group differences)
  2. what about contextual effect (the effect of the group level characteristics on individual outcomes after controlling for individual level confounders)
  3. what about cross-level interaction (specific combination of individual and contextual characteristics)

Multilevel analysis is the most suitable model to investigate the effect of ecological factors on individual's outcomes. I was wondering why authors did not consider it, given the availability of such rich data?

Round 2

Reviewer 3 Report

The statement in page 8, lines 276-280 regarding multilevel analysis is not correct:

“The use of bootstrap weights to account for survey design effects is a major strength of the analysis, since the weighting procedure ensures estimates are accurately representative of the Canadian household population. However, weighting also limits the ability to use a multilevel model which would account for compositional and contextual effects in addition to cross-level interactions”.

In fact, it is possible to fit a multilevel model in complex survey data with design weights. It is more appropriate to discuss this issue in the limitation section by saying, e.g. this study focuses only on the fixed effects and future studies should take both fixed and random effects into consideration…
